# Lowering the barriers to sexual health services: Impacts of free counselling and testing for sexually transmitted infections in Switzerland – an observational study

Leonie Arns-Glaser[1]*, Andrea Farnham[1], Katja Hochstrasser[2], Jan S. Fehr[1], Benjamin Hampel[1,3]

1 Department of Public Health & Global Health, Epidemiology, Biostatistics and Prevention Institute, University of Zurich, Zurich, Switzerland, 2 SpiZ Sexualpädagogik in Zürich, Zurich, Switzerland, 3 Checkpoint Zurich, Zurich, Switzerland

* leonie.arns-glaser@uzh.ch

## Abstract

### Background

In Switzerland, tests for HIV and sexually transmitted infections (STIs) are usually not covered by health insurance in asymptomatic people. To improve access, Zurich launched free voluntary counselling and testing (VCT) in June 2023 for residents <26 years or with low income. This study describes the implementation of free VCT for HIV and STIs in a high-income setting where access to testing was previously expensive, along with key barriers and enablers to accessing testing and counselling in the target population.

### Methods

We conducted a study using routine health data, and a client feedback questionnaire (FBQ) collected during the first 12 months of the programme. Logistic regression models were used to assess factors associated with first-time HIV or STI testing, with results reported as odds ratios (ORs) and 95% confidence intervals (CIs).

### Results

In the first year, 3,475 people came for free testing. 83% (n = 2,866) agreed to share their data. 21% (n = 719) completed the FBQ. Median (IQR) age of participants was 24 (23, 26) years. 46% were assigned female at birth. Four HIV diagnoses were confirmed, all of them in MSM. The infection with the highest positivity rate was chlamydia (4.5%), followed by gonorrhoea (2.5%). Men having sex with men (MSM) showed the highest positivity rate in all STIs. 39% of visits were by individuals who had not received prior HIV testing. MSM were significantly less likely to be first-time testers for HIV (OR: 0.28, 95% CI: 0.15–0.48) and also less likely to be first-time

**Data availability statement:** Datasets analysed during the current study and used to generate tables, figures, and the supplementary material are not publicly available due to the sensitive nature of the data yielded by this highly representative, individual-level dataset. Source data are thus not provided with this paper. Investigators with a request for selected data should send a proposal to the SwissPrEPared e-mail address (info@swissprepared.ch). The provision of data will be considered by the Scientific Board of the SwissPrEPared cohort study and the relevant study team. Data provision is subject to Swiss legal and ethical regulations and will be detailed in a material and data transfer agreement.

**Funding:** This study was financially supported by the City of Zurich (City Council Resolution) in the form of a grant (2021/432) received by JSF. This study was also financially supported by the Swiss Federal Office of Public Health (FOPH) through support to the SwissPrEPared program and study in the form of an award received by JSF. No additional external funding was received for this study. The funders had no role in study design, data collection and analysis, decision to publish, or preparation of the manuscript.

**Competing interests:** The authors have read the journal's policy and have the following competing interests: BH reports honoraria for advisory boards, lectures, and travel grants paid to himself from Gilead, MSD, and ViiV Healthcare, which are unrelated to the submitted work. JSF reports grants to the institution from Gilead, MSD, and ViiV Healthcare, which are unrelated to the submitted work. None of the authors are employed by these companies. The other authors declare no competing interests. This does not alter our adherence to PLOS ONE policies on sharing data and materials.

testers for STIs (OR: 0.74, 95% CI: 0.44–1.21) compared to women who have sex with men (WSM); however, the latter association was non-significant.

## Conclusions

The free VCT project experienced high demand during the first year. Even in a high-income setting, counselling improved participants' sexual health knowledge and facilitated many first HIV/STI tests.

---

## Background

Globally raising rates of sexually transmitted infections (STIs) highlight a growing sexual health burden, and Switzerland reflects these trends [1–3]. Young people in particular experience a disproportionate burden of STIs, with the canton of Zurich reports the highest national incidence. The proportion of adolescents and young adults engaging in condomless sex has also increased in recent years, underscoring the need for accessible HIV/STI testing and counselling services [2–6].

Parallel to these STI trends, recent HIV surveillance indicates ongoing disparities in Switzerland. Although HIV diagnoses have declined following a temporary post-COVID rise, important differences persist across populations and regions. In 2023 352 people received a new HIV diagnosis followed and 318 in 2024, with men who have sex with men (MSM) accounting for most new infections in men in both years. MSM remain disproportionately affected compared to heterosexual populations and are diagnosed at younger ages. Regional differences are marked, with the highest incidence observed in Zurich and the Lake Geneva region. While the national HIV pre-exposure prophylaxis (PrEP) programme (SwissPrEPared) has likely contributed to a faster decline in new infections among MSM, this group continues to represent a key population for HIV prevention [2,3].

Voluntary counselling and testing (VCT) is an effective strategy to promote safer sexual practices, support early diagnosis, and reduce STI transmission, particularly among young people [7–15]. However, in Switzerland, HIV and asymptomatic STI tests are not consistently covered by health insurance, and high deductibles frequently shift costs to individuals. Financial barriers have been repeatedly identified as major deterrents to testing in both high-income and lower-income settings [16–18].

To address these barriers, the city of Zurich launched a pilot programme in June 2023 offering free HIV and STI counselling and testing to residents under 26 years of age or with low income [19,20]. Despite widespread availability of subsidised or low-cost testing in many high-income settings, there is limited evidence on how free testing affects uptake, barriers, and user characteristics, particularly in real-world European contexts. Moreover, little is known about whether such programmes increase service workload in existing sexual health structures.

This study aimed to describe the implementation and uptake of free VCT for HIV and STIs in a high-income setting where testing was previously costly. Specifically, we examined (i) the advantages of providing free HIV/STI testing and counselling,

(ii) barriers and enablers reported by participants, and (iii) the potential added burden on existing sexual healthcare services resulting from the implementation of free VCT.

## Methods

### Study setting

The city of Zurich began a three-year pilot project (1 June 2023–31 May 2026) aimed at providing free VCT. For this analysis, we included data from visits between 1 June 2023 and 31 May 2024. Individuals are eligible to participate if they live in the city of Zurich and are either under 26 years of age or take part in a social discount programme which identifies them as low-income ("KulturLegi"). Participants were eligible to receive free testing for HIV, syphilis, *Chlamydia trachomatis* (CT) and *Neisseria gonorrhoea* (NG), and Hepatitis C (HCV), depending on the shared decision between the participant and the healthcare professional (HCP) during the counselling (see S1 File for a description of the laboratory tests). The free testing could either be accessed in the form of classical VCT or in the form of a PrEP visit. Two VCT sites run by the "Sexuelle Gesundheit Zurich (SeGZ)" were commissioned to run the intervention: Checkpoint Zurich, the larger of the two centres has traditionally focused on serving the queer community. In contrast, the Test-In has placed greater emphasis on catering to the general public.

### Data collection

This cross-sectional study consists of three different quantitative data sets. The first two datasets are routine VCT data among the general population (BerDa data set) and the population taking PrEP in Switzerland (SwissPrEPared data set), respectively. This includes operational data (e.g., number of visits), laboratory results (e.g., positive test results), and personal characteristics and self-reported data of those receiving VCT (e.g., demographics, socio-behavioural data, sexual behaviours). In addition to these routinely collected data, the third dataset consisted of an anonymous follow-up survey that was sent digitally to participants' phones via text message after their free VCT visit (FBQ data set) (excluding people who took part in the pilot as part of a PrEP visit, due to different administrative processes). The interactions of the three data sets are depicted in Fig 1 and described in more detail in the S2 File.

### Statistical analysis

Participants ≥ 26 years of age were considered low-income, according to the inclusion criteria. People were categorised according to their gender and sexual orientation (see S3 File). Women identifying as heterosexual and bisexual were grouped together to women having sex with men (WSM), as they have similar STI risks. Similarly, men identifying as bisexual, homosexual or pansexual were grouped together as men having sex with men (MSM). Men identifying as heterosexual were considered to be men exclusively having sex with women (MSW) and women identifying as homosexual were considered to be women exclusively having sex with women (WSW).

We used median and IQR for continuous non-normally distributed variables, mean and SD for continuous normally distributed variables and total (percentages) for categorical variables. Normality of continuous variables was assessed with the Anderson-Darling test. To examine associations between first-time HIV or STI testing (binary outcomes) and selected covariates, we fitted logistic regression models. Covariates included age, low-income qualification, sex, country of birth (Switzerland or Liechtenstein vs. elsewhere), educational attainment (university degree vs. not), and sexual identity group (WSM vs MSW, and MSM, and others).

Data analysis was performed in R (version 4.4.0) using the dplyr, ggplot2, HH, Hmisc, and rstatix package [21–25].

### Outcomes

Although PrEP users have internal programme identifiers, and repeat visits can be tracked, this cannot be done for VCT visits, as the free VCT programme operates anonymously. An individual may therefore contribute multiple visits if they

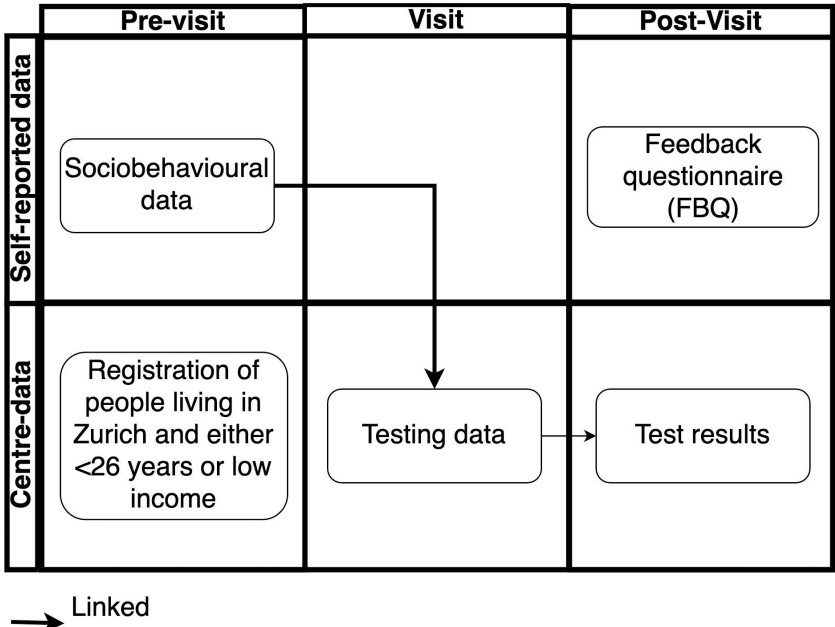

**Fig 1. Study flow chart.**

returned for repeat testing, but unique identifiers are not collected and repeat attendance cannot be reliably distinguished. Visits are therefore our unit of analysis.

The primary outcomes of the study were: (1) self-reported perceived benefits and hurdles to accessing voluntary counselling and testing (VCT); (2) the number of new HIV and STI diagnoses over a 12-month follow-up period; and (3) first-time HIV and STI testing, including associated sociodemographic and behavioural correlates identified through logistic regression analysis. Secondary outcomes included demand on existing sexual healthcare centres, measured by the number of overall visits and tests. Tests for HIV, HCV, and syphilis were considered positive only after confirmatory testing.

## Ethical evaluation

In accordance with the declaration of Helsinki, the evaluation of the pilot project has been exempted from ethical approval under the Swiss Human Research Act (HFG) by the Cantonal Ethics Committee of Zurich (BASEC register number: Req-2023–00594) as data were collected anonymously. Ethical approval for the SwissPrEPared study was received by all relevant local ethics committees (lead committee: Zurich, Switzerland—BASEC registration number: 2018–02015) on 15 May 2019 and was registered with ClinicalTrials.gov (NCT03893188) on 14 March 2019.

For non-operational data, we only included information from individuals who explicitly agreed to share their data for research purposes, either when filling in the electronic BerDa form during their VCT visits by choosing between not sharing any data, sharing minimal data or all data for scientific use in the online form (anonymous data), or by consenting to participate in the SwissPrEPared study during a PrEP visit by giving written informed consent to be part of the SwissPrEPared study together with a HCP. People between 16–18 years were eligible but in the case of participation via SwissPrEPared, they required parental consent.

## Results

### Demographics of people accessing free HIV and STI VCT

Overall, 3,475 free VCT were carried out in the 12-month period between June 1, 2023 and May 31, 2024. 3,274 (94%) were regular VCT visits and 201 (6%) were PrEP visits. In 2,866 visits (83%), the participants agreed to make their socio-behavioural data and test results available for the scientific evaluation.

The proportion of visits from people qualifying due to low-income was overall 16% (N = 467) of all study visits. The remaining 84% of visits were from participants under age 26 (see Table 1). 67% (N = 1,725) visits were by people born in Switzerland or Lichtenstein, followed by Germany (9%, N = 308), Italy (2%, N = 58), France (2%, N = 47), United States of America (1%, N = 35), and Austria (1%, N = 33).

1,378 (48%) visits were by participants assigned male at birth and 1,180 (41%) by participants assigned female at birth (see Table 1). 14 visits were by transgender women (0.5% of total) and 12 by transgender men (0.4% of total). The overall median age (IQR) was 24 (23, 26) years (see Table 1 and S4 Fig.). People qualifying by their age (N = 2,096) were a median of 23 years (IQR: 22, 25) years. People qualifying due to low-income were a median of 30 years (IQR: 28, 36) years. 4% (N = 116) were below the age of 20. Approximately 43% (N = 1,231/2,866) reported to be WSM, 29% (N = 830/2,866) MSW, 24% (N = 683/2,866) as MSM, 3% (N = 97/2,866) as gender diverse and 1% (N = 24/2,866) as WSW (see Table 1). These proportions were similar between the low-income and young groups.

The largest number of visits were by participants who were tested as part of a routine checkup (33%, N = 836), closely followed by insecurity after sexual contact (26%, N = 662). The reasons for accessing testing were similar across the different sociodemographic groups, with the exception of obtaining PrEP, which was the second most cited reason for testing among the MSM population (N = 161, 27%), compared to N = 193 (7%) in the overall study population (see Table 1).

**Table 1. Demographic characteristics and reasons for accessing VCT stratified by sex assigned at birth and population groups.**

| | Overall | Male sex at birth | Female sex at birth | No sex given[a] | Women having sex with men | Men having exclusively sex with women | Men having sex with men | Women having exclusively sex with women | Gender diverse |
|---|---|---|---|---|---|---|---|---|---|
| N | 2866 | 1378 | 1180 | 308 | 1231 | 830 | 683 | 24 | 97 |
| Percentage (%) | – | 48% | 41% | 11% | 43% | 29% | 24% | 1% | 3% |
| Median (IQR) age | 24 (23, 26) | 24 (23, 26) | 24 (23, 26) | 24 (23. 26) | 24 (23, 26) | 24 (23, 26) | 24 (23, 26) | 24 (22.75, 25) | 24 (22, 25) |
| Mean ± sd age | 25.0 ± 4.9 | 25.2 ± 5.2 | 24.7 ± 4.5 | 25.1 ± 5.05 | 24.8 ± 4.5 | 24.8 ± 4.2 | 25.6 ± 6.2 | 23.6 ± 2.0 | 25.1 ± 5.7 |
| Age range | 15 - 68 | 16 - 68 | 15 - 67 | 17-58 | 15 - 67 | 17 - 63 | 16 - 68 | 17 - 26 | 16 - 52 |
| Low income N (%) | 467 (16%) | 228 (17%) | 184 (16%) | 55 (18%) | 194 (16%) | 125 (15%) | 129 (19%) | 0 (-) | 18 (19%) |
| Insecurity after sexual contact | 662 (26%) | 284 (21%) | 378 (32%) | NA | 356 (33%) | 200 (27%) | 79 (13%) | 2 (10%) | 16 (19%) |
| New relationship | 499 (20%) | 253 (18%) | 246 (21%) | NA | 232 (21%) | 209 (28%) | 44 (7%) | 4 (20%) | 10 (12%) |
| Routine examination | 836 (33%) | 442 (32%) | 394 (34%) | NA | 366 (33%) | 225 (31%) | 201 (34%) | 6 (30%) | 38 (45%) |
| PrEP | 193 (7%) | 177 (13%) | 16 (1%) | NA | 12 (1%) | 6 (1%) | 161 (27%) | 2 (10%) [b] | 12 (14%) |
| Symptoms | 103 (4%) | 72 (5%) | 31 (3%) | NA | 29 (3%) | 33 (5%) | 37 (6%) | 1 (5%) | 3 (4%) |
| Confirmation test | 13 (1%) | 7 (1%) | 6 (1%) | NA | 5 (0%) | 3 (0%) | 4 (1%) | 1 (5%) | 0 (-) |
| Sex outside of relationship | 66 (3%) | 30 (2%) | 36 (3%) | NA | 34 (3%) | 13 (2%) | 14 (2%) | 1 (5%) | 4 (5%) |
| Partner or self-diagnosed with STI | 85 (3%) | 52 (4%) | 33 (3%) | NA | 30 (3%) | 22 (3%) | 30 (5%) | 2 (10%) | 1 (1%) |
| Partner HIV acquisition | 2 (0%) | 2 (0%) | 0 (-) | NA | 0 (-) | 1 (0%) | 1 (0%) | 0 (-) | 0 (-) |
| Other | 80 (3%) | 50 (4%) | 30 (3%) | NA | 29 (3%) | 25 (3%) | 24 (4%) | 1 (5%) | 1 (1%) |

[a]In N = 308 only minimal consent was given (e.g., age and test results), [b]2 visits by a transgender women identifying as homosexual

## Benefits of free VCT: results of testing

39% (N = 986/2,557) of visits were from people receiving an HIV test for the first time. This number was lower among those who were eligible due to low-income (23%, N = 94) compared to those eligible due to young age (< 26 years) (42% N = 892).

Four people received a confirmed HIV diagnosis during the first study year. The STI with the highest positivity rate was CT (4.5% positive, N = 142/3,181), followed by NG (2.8% positive, N = 90/3,181). Two acute and 9 latent syphilis diagnoses were found within the project of the 2,938 tests performed. There were no confirmed positive cases of HCV during the study period among the 1,067 tests performed (see Table 2).

The leading STIs among those assigned male at birth were CT and NG with a positivity rate of 4.4% and 3.9%, respectively. The leading STI among those assigned female at birth was CT (positivity rate 4.5%). 9 out of 1,112 NG tests in individuals assigned the female sex at birth were positive (positivity rate 0.8%). The demographic group of MSM carry proportionally the highest burden of reactive tests overall (3.5%) and for each pathogen, with positivity rates of 5.8% for CT and 7.7% for NG. Moreover, all 4 confirmed HIV cases were in MSM, as well as the only stage 1 syphilis case and 5 of the 6 latent syphilis cases (see Table 2).

**Table 2. HIV/STI-testing results stratified by sex assigned at birth and population groups.**

| | Overall[a] | Male sex at birth | Female sex at birth | No sex given[b] | Women having sex with men | Men having exclusively sex with women | Men having sex with men | Women having exclusively sex with women | Gender diverse |
|---|---|---|---|---|---|---|---|---|---|
| N | 2866 | 1378 | 1180 | 308 | 1231 | 830 | 683 | 24 | 97 |
| Percentage (%) | – | 48% | 41% | 11% | 43% | 29% | 24% | 1% | 3% |
| Testing | | | | | | | | | |
| | | | | | | | | | |
| Chlamydia | | | | | | | | | |
| Performed, N | 2642 | 1248 | 1112 | 283 | 1148 | 779 | 599 | 22 | 94 |
| Positive, N | 119 | 55 | 50 | 14 | 54 | 28 | 35 | 0 | 2 |
| Positivity rate | 4.5% | 4.4% | 4.5% | 5.0% | 4.7% | 3.6% | 5.8% | 0 | 2.1% |
| Gonorrhoea | | | | | | | | | |
| Performed, N | 2642 | 1248 | 1112 | 283 | 1148 | 779 | 599 | 22 | 94 |
| Positive, N | 66 | 49 | 9 | 8 | 12 | 6 | 46 | 0 | 2 |
| Positivity rate | 2.5% | 3.9% | 0.8% | 2.6% | 1.1% | 0.8% | 7.7% | 0 | 2.1% |
| HIV | | | | | | | | | |
| Performed, N | 2551 | 1247 | 1043 | 261 | 1082 | 757 | 607 | 18 | 87 |
| Confirmed acquisition, N | 4 | 4 | 0 | 0 | 0 | 0 | 4 | 0 | 0 |
| Hepatitis C (HCV) | | | | | | | | | |
| Performed, N | 878 | 439 | 361 | 78 | 377 | 262 | 206 | 5 | 28 |
| Positive, N | 0 | 0 | 0 | 0 | 0 | 0 | 0 | 0 | 0 |
| Syphilis | | | | | | | | | |
| Performed, N | 2415 | 1168 | 988 | 259 | 1019 | 707 | 582 | 20 | 87 |
| Early stage[c], N | 1 | 1 | 0 | 0 | 0 | 0 | 1 | 0 | 0 |
| Late Stage[d], N | 6 | 5 | 0 | 1 | 0 | 0 | 5 | 0 | 1 |

[a]Testing numbers from the overall column are derived from operative data [b]In N = 308 only minimal consent was given (e.g., age and test results), [c]Stage I/II, [d]Stage III/latent

Among visits from individuals < 26 years, CT had the highest positivity rate at 5.0%, compared to a lower CT positivity rate of 2.7% in visits of low-income participants. All confirmed HIV acquisitions were found in visits from participants < 26 years of age.

## Benefits of free VCT: Feedback from participants

The FBQ with feedback on their visit was filled out after 22% (N = 719/3274) of the regular VCT visits. Those who responded to the FBQ after their visit did not differ in age from the general participant population (see Table 3 and S4 Fig.). Nonetheless, they were more likely to be assigned female at birth (58%, N = 419/717 vs. 43% in the general participant population). The biggest group filling in the FBQ were WSM (N = 386/719, 53%). Moreover, 3 transgender women and 3 transgender men filled in the FBQ. 19% (N = 128/682) of responses to the FBQ come from people qualifying due to low-income (see Table 3). The demographic and sexual identity groups did vary slightly between those qualifying by age and those by low-income. A high proportion of respondents in both groups had a university degree or were currently pursuing higher education. Among respondents under 26 years of age, 49% (N = 270) and 80% (N = 438) reported having or working toward a degree, out of whom N = 220 fall into two categories, holding a degree while continuing their studies. Similarly, among respondents qualifying due to low-income, 77% (N = 98) reported having a university degree and 45% (N = 57) were in some form of education. Of these 44 respondents fell into both categories, holding a degree while continuing their studies (see S5 Table).

The proportion of first-time testers among those who filled out the FBQ was similar to that of the overall study population: 39% (N = 276/710) had no prior HIV test and 39% (N = 278/710) came for their first STI test. This again varied depending on whether participants qualified by age or income. 44% (N = 243/549) of those < 26 years had no prior HIV test versus 15% (N = 19/124) of low-income participants, while 43% (N = 236/546) of those aged < 26 years had no prior STI test versus 21% (N = 27/128) of low-income participants. Logistic regression models were used to assess factors associated with having previously received HIV or STI testing, as reported in the FBQ (see Fig 2, S6 Table). For first-time HIV testing, significant associations were observed for sexual identity and birthplace. Compared WSM, MSM had lower odds of being first-time HIV testers (OR: 0.28, 95% CI: 0.15–0.48), as did MSW (OR: 0.60, 95% CI: 0.40–0.88). Participants born outside Switzerland were more likely to be first-time HIV testers than those born in Switzerland (OR: 1.49, 95% CI: 1.01–2.20).

For first-time STI testing, several factors were associated with testing status. Participants who qualified based on low income had lower odds of being first-time STI testers compared to those who qualified by age (OR: 0.71, 95% CI: 0.36–1.40). MSM had lower odds compared to WSM (OR: 0.74, 95% CI: 0.44–1.21), while MSW had higher odds (OR: 1.46, 95% CI: 1.01–2.13) of being first-time STI testers. Moreover, increasing age was associated with lower odds of being a first-time tester (OR per year: 0.92, 95% CI: 0.86–0.98) (see Fig 2, S6 Table).

**Table 3. Demographics of the feedback questionnaire stratified by population groups.**

|  | Overall | Women having sex with men | Men having exclusively sex with men | Men having sex with men | Women having exclusively sex with women | Gender diverse | other |
|---|---|---|---|---|---|---|---|
| N | 719 | 386 | 186 | 103 | 3 | 28 | 13 |
| Percentage (%) | – | 54% | 26% | 14% | 0% | 4% | 2% |
| Median (IQR) age | 24 (23, 26) | 24 (23, 26) | 24 (23, 26) | 25 (23, 26.75) | 24 (24, 24.5) | 24 (22, 25) | 25.5 (24, 27) |
| Mean ± sd age | 25.3 ± 5.2 | 25.0 ± 4.6 | 25.0 ± 4.1 | 26.8 ± 8.0 | 24.3 ± 0.6 | 24.5 ± 4.2 | 26.4 ± 5.4 |
| Age range | 16 - 70 | 17 - 54 | 18 - 52 | 16 − 70 | 24 - 25 | 19 - 39 | 18 - 40 |
| Low income N (%) | 128 (18%) | 66 (17%) | 30 (16%) | 25 (24%) | 0 (-) | 4 (14%) | 3 (23%) |

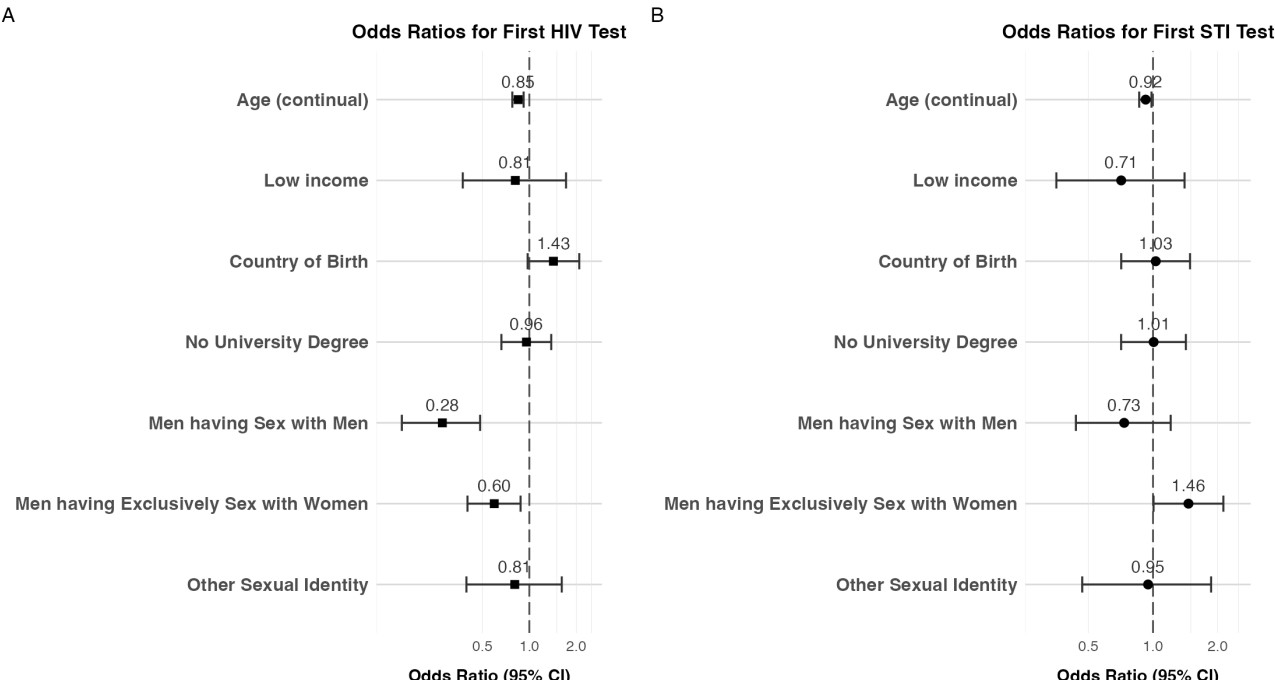

**Fig 2. Forest plots showing odds ratios with 95% confidence intervals for factors associated with first-time HIV testing (Panel A) and first-time STI testing (Panel B) among survey participants.** Odds ratios greater than 1 indicate higher odds of being a first-time tester, while odds ratios less than 1 indicate lower odds. Age is presented as a continuous variable, with odds ratio reflecting the change in odds per one-year increase in age. For categorical variables, reference groups are: participants qualifying by age (for Low income), born in Switzerland (for Country of Birth), university degree (for No University Degree), and women having sex with men (for sexual identity categories). Models were adjusted for all variables shown. Data are from the feedback questionnaire (FBQ) (N = 719), with 48-50 observations excluded due to missing values.

The benefits of the VCT appointment as reported by those who completed the FBQ were: knowing their STI status (89%, N = 637/719), the possibility to have a non-judgmental conversation on their sexuality (56%, N = 400/719), the possibility to receive answers to one's questions on sexual health (34%, N = 242/719), and receiving knowledge on STI- and HIV-prevention (26%, N = 185/719) (see Fig 3A). In the free text, the responders to the FBQ emphasised their appreciation for being able to talk about polygamous relationships, their gender identity, recommended vaccinations as well as the openness of the counsellors.

80% (N = 568/714) of participants agreed completely or partly that they learned something about their sexual health during the VCT appointment. Participants overwhelmingly reported positive experiences with the counselling, with 97–99% agreeing completely or partially across various measures, such as trust in the counsellor and understanding the reasons for testing (details in S7 Table).

In addition to concerns about their sexual health, 13% (N = 90/683) indicated having other health concerns. 70% (N = 63/90) of participants who indicated that they had other health concerns, reported that they were worried about their mental health, and 14% (N = 13/90) about substance use. 13% (N = 12/90) were worried about general medical issues, but did not have a general practitioner they could consult. 23% (N = 21/90) reported that their health insurance deductible was too high, which prevented them from visiting a general practitioner about their medical problems.

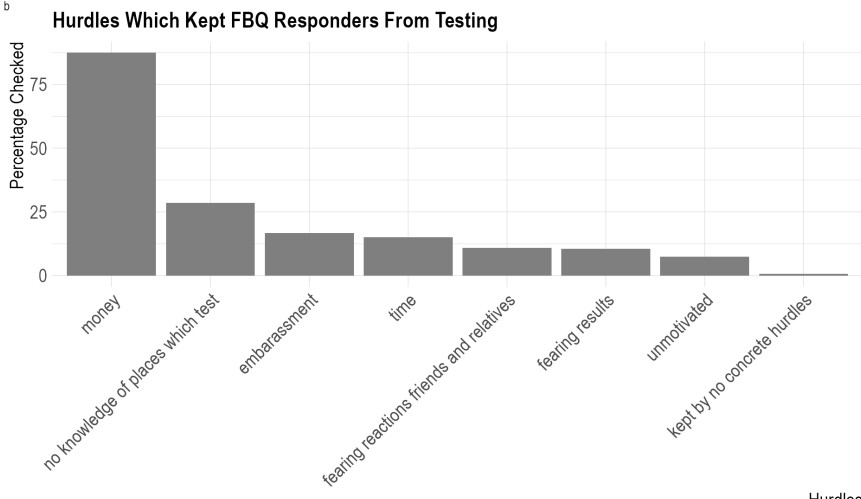

**Fig 3. (Panel A) Perceived benefits and (Panel B) barriers to STI among feedback questionnaire respondents.**

### Key barriers to accessing testing

58% (N = 407/719) of the FBQ answers indicated having experienced barriers to HIV- and STI testing in the past. Of those, 88% (N = 356/407) indicated that it was due to financial reasons. (The second most common barrier was "no knowledge of places that test" (N = 116/407, 29%) (see Fig 3B). Additional hurdles that were mentioned in the free text emphasised the financial aspect (N = 12/33), issues with booking the appointment or general accessibility (N = 9/33), that previously visited HCPs were unwilling or unable to perform the requested tests (N = 7/33), being afraid of the procedure (N = 5/33), being unclear about their own risk (N = 5/33), issues related to insurance (N = 3/33), issues related to

practitioners not being versed to consult and treat trans-people (N = 2/33), and being afraid of being judged by the medical professionals (N = 2/33).

Additionally, 347 participants (49%) indicated that, without the intervention, they would have gotten tested less frequently, while 240 (34%) stated they would not have gotten tested at all. Participants would have been willing to get tested for a median (IQR) price of 50 (30, 50) Swiss Francs, corresponding to 58 (35, 58) US Dollars and 53 (32, 53) Euros during the study period (N = 561 participants answered this question).

### Indicators of added burden on existing healthcare structures

During the 12-month study period, a total of 3,475 visits were conducted under the free VCT programme. In this period, 3,079 HIV-tests, 3,181 CT- and NG-tests, 2,938 syphilis-tests, and 1,067 HCV tests were carried out. Among FBQ respondents, 163 (23%) could not book an appointment at the time they wanted, and these participants reported a median wait of 25 days (IQR: 14.5, 30 days) longer than desired.

## Discussion

This observational study aimed to evaluate the implementation of free VCT for HIV and STIs among young and low-income people in Zurich, Switzerland, a high-income setting where testing outside of the pilot project is expensive and typically paid for out-of-pocket. Our findings highlight several benefits of the programme, including increased accessibility and uptake of testing among young and low-income populations (49% of FBQ respondents indicated that they would have tested less regularly and 34% would not have gotten tested at all). Participants identified both structural and individual barriers to accessing testing, including costs, stigma, and logistical challenges. While some experienced long waits (a median delay of 25 days among 23% of respondents), the overall burden on the sexual healthcare centres was manageable. These findings imply that even in high-income settings the financial burden of accessing testing can be significant. To our knowledge, this is one of the first evaluations of a free testing programme in a high-income country, offering insights into how such initiatives can improve access while also providing initial indicators of their potential implications for sexual healthcare service capacity.

### Benefits of implementing free counselling and testing initiative

Participants rated the counselling sessions highly, with 80% reporting they learned something about their sexual health and >97% describing their interactions with the counsellors as positive (see S7 Table). These findings align with studies highlighting the benefits of VCT, particularly for young people [11–15]. Notably, 34% identified learning about sexual health as a direct benefit, underscoring the importance of expert advice even in an era of easily accessible information. Personal counselling has been shown in another Swiss study to play a critical role in encouraging STI testing [26], and face-to-face discussions remain valuable, as young people often prefer them over online counselling in sexual health and other medical contexts [27,28]. Reviews also indicate that VCT can reduce HIV risk behaviours [8,29], reinforcing the need to expand access to VCT for young people.

The percentage of first-time testers and positivity rates of HIV and STIs varied between sociodemographic groups. The positivity rate for STIs overall was highest in MSM, their odds of having received prior STI testing were also the highest. However, the OR was not significant. MSM as well as MSW have higher odds to have received HIV testing than WSM. However, MSW have lower odds of having been tested for STIs before than WSM.

The positivity rates of the HIV and STI testing are comparable to the literature [2,3,26,30,31]. Population-level reductions in HIV and STI incidence will only potentially be seen in the longer term. Nonetheless, considering the high burden of HIV infections in MSM compared to the general public, providing free HIV testing in MSM is probably cost-effective [2,3,32,33].

## Key barriers to accessing testing

The majority (58%) of FBQ respondents reported encountered barriers to testing, with cost being the most cited (88% of respondents). This aligns with previous findings, such as the popularity of low-cost VCT in Bern [26] and free CT screening in the southwest of Switzerland [30]. Moreover, FBQ responders indicated other barriers related to the Swiss healthcare system, such as lack of a general physician or high health insurance deductibles. In 2004, Swiss residents paid 25% of all health-costs out-of-pocket, which is higher than the OECD (Organisation for Economic Co-operation and Development) average of 19%. Moreover, 17% of Swiss residents had high out-of-pocket spending (i.e., spending 10% of income on out-of-pocket health costs, 5% if affected by poverty) and were thus characterised as underinsured [34]. The next most frequently cited barrier by the FBQ responders was lack of knowledge on where to be tested (29% of respondents). Logistical issues, such as limited clinic hours and long waiting times, also posed challenges.

The study also identified other key gaps in the healthcare of the target population, with 13% of the respondents reporting other concerns about their health beyond sexual health. These were mostly centred around mental health and substance use. The identification of these gaps highlights the need for a more holistic approach to healthcare delivery, where sexual health services are integrated with broader support for other inter-related medical needs.

It is surprising that people qualifying for the programme due to low income are relatively young (median 30 years old), only six years older than those qualifying by age. This may suggest that people in younger age groups tend to have more sexual partners and thus have an increased risk for STIs. However, it may also suggest that the programme is not reaching older populations. Previous work has shown that people with insufficient local language knowledge or ability to navigate digital content do not receive important information on the qualifying social discount programme (KulturLegi). Moreover, the process of registering or renewing social discount programme membership was complicated and participants indicated that they would not have managed without help [35]. This suggests that low-income individuals who are relatively well-educated or with good access to information may be better positioned to navigate and benefit from the social discount programme, such as students > 26 years pursuing their master's degree or doctoral studies. This is supported by the fact that 77% of > 26 years old FBQ responders have at least a Bachelor's degree and 45% are currently pursuing an education, which might further explain the relatively young median age of low-income people of 30 (28–36) years. Targeted outreach and education efforts in partnership with community organisations, such as community workshops or peer-led initiatives, could help bridge the gap and facilitate access to VCT for low-income populations with varying levels of health and digital literacy.

In addition to these observations, the relatively low number of PrEP-related visits in our dataset can be attributed to structural characteristics of the programme. For example, the national SwissPrEPared cohort study indicates that PrEP users in Switzerland are typically older (median age 40 years; IQR 33–47) and tend to be highly educated and financially secure [36]. Similar age distributions have been reported in other European PrEP initiatives: participants had median ages around 35 years in the UK PROUD trial and the ANRS IPERGAY study (France and Canada), and approximately 40 years in the Netherlands' AMPrEP project [37–40]. These characteristics differ substantially from the target population of the free VCT programme, which primarily consists of young people and individuals with low income. As a result, most PrEP users do not qualify for the free VCT offer, which explains their limited representation in our data.

Only 5% (n = 116) of participants qualifying by their age were younger than 20 years old. It is unclear whether adolescent participants are not coming because they do not yet need VCT, because they are unaware of the programme, or whether they are unaware that they are at risk of STIs. A Swiss national survey from 2017 has found the average age of the first sexual contact to be 16.7 ± .05 years and a 2024 WHO report stated that 13% of girls and 17% of boys at the age of 15 years have had sexual intercourse in Switzerland [41,42]. In the USA, the percentage of 14–18-year-olds who ever had sex dropped between 1991–2019 from 54% to 38% [42]. Hence, it might be argued that many Swiss adolescents do not need STI testing. However, young people could particularly benefit from sexual health counselling, as this can lead to

changes in sexual risk behaviour [8]. Further research is needed to assess if there is no need in younger people or if more outreach in these population groups is needed.

### Indicator of added burden on existing healthcare structures

These indicators do not allow for a direct comparison with pre-programme workload, because the implementation of free VCT was accompanied by additional staffing. However, the proportion of clients who could not obtain an appointment at their preferred time suggests that demand may have exceeded available capacity during certain periods.

### Strengths and limitations

A strength of our study is its comprehensive approach, going beyond prevalence and socio-behavioural data to include participants' subjective assessments of the benefits and accessibility assessment of the VCT programme. Additionally, 83% of all participants shared their data, and the sample includes people from diverse socio-demographic backgrounds, genders and sexual orientations. Moreover, the study captures not only epidemiological data but also insights into barriers to testing, communication during the VCT, and perceived benefits from the perspective of the participants.

There are a few limitations to our study: Firstly, due to the anonymous nature of the testing, we cannot determine whether an individual contributed more than one VCT visit, nor can we link multiple FBQ responses that may come from the same person. In addition, VCT visit data and FBQ responses cannot be linked to one another at the individual level. For these reasons, all analyses are performed at the level of visits and completed questionnaires rather than unique persons. Importantly, any repeat attendance would bias the proportion of first-time testers downwards, because repeat visitors are, by definition, not first-time testers. The reported 39% first-time testing rate at the visit level is therefore a conservative estimate; the true proportion at the individual level is likely higher.

The FBQ provides a generally representative overview of the VCT population, with a modest overrepresentation of individuals identifying as WSM, consistent with typical survey response patterns observed in the literature (51). Participation was voluntary, which may introduce some selection bias, though this is a common feature in survey-based research. A limitation is that individuals participating via PrEP visits—predominantly MSM (83%)—were not invited to complete the FBQ due to differing administrative processes. However, this group represents approximately 7% of the overall population, so the impact on overall representativeness is limited. Nevertheless, this may lead to a slight underrepresentation of MSM perspectives and a relative emphasis on the experiences of WSM. Because additional staff positions were created specifically for the free VCT programme, we were unable to compare workload directly with the pre-programme period. A true baseline-versus-programme capacity analysis was therefore not possible, and our assessment of burden relies only on indirect indicators such as appointment availability and patient-reported waiting times. These indicators should therefore be interpreted cautiously and not as definitive measures of clinical workload or capacity.

### Conclusions

The pilot project offering free HIV and STI VCT demonstrates that individuals are highly likely to utilise these services when they are readily available and free of charge. However, the findings highlight the need for additional resources to sustain and expand the programme. Efforts must be intensified to reach populations with lower educational backgrounds, as they are currently underrepresented among participants.

Among all population groups, MSM have the greatest epidemiological benefit from receiving testing, as reflected by the four confirmed HIV diagnoses and the overall highest positivity rate of all tests within this group. Hence, a key challenge remains in reaching MSM who do not currently access testing services. Future efforts should focus on improving outreach to this population, perhaps through community-based campaigns and mobile testing units, maximising the public health impact of the programme. The results clearly show the advantages to individual sexual and mental health and well-being of knowing one's health status and providing access to expert counselling. In future projects introducing free VCT, it is crucial to allocate additional resources to outreach programmes to target people with lower health literacy and less informed social networks.

## Supporting information

**S1 File. Description of the laboratory tests performed for STI / HIV diagnosis.**
(PDF)

**S2 File. Description of the data sets.**
(PDF)

**S3 File. Definitions of demographic and sexual identity groups.**
(PDF)

**S4 Fig. Age distribution of the feedback questionnaire (FBQ) responders and participants of the intervention (study population).**
(TIF)

**S5 Table. Descriptives of people < 26 years of age and people on low incomes filling in the feedback questionnaire (FBQ).** [1]Entries with age provided, CH-Li: Switzerland or Lichtenstein.
(PDF)

**S6 Table. Odds ratios (95% CI) from logistic regression on first HIV/STI tests, feedback questionnaire (FBQ) data.** [1]Other: consisting of gender diverse people, women having exclusively sex with other women, and participants not using terms for either their gender or sexual orientation identity. [2] Emphasises specific sexual behaviour patterns or identity, transcending binary orientation categories. Abbreviations: 95% CI: 95% confidence interval for point estimate Lower being the lower limit and upper the upper limit of the 95% CI, HIV: human immunodeficiency virus, model: first test ~ low income + age + level of education + country of birth + demographic group, OR: odds ratio, STI: sexually transmitted infection.
(PDF)

**S7 Table. Likert-Matrix on the communication during the consultation as recorded in the feedback questionnaire (FBQ).**
(PDF)

## Acknowledgments

We sincerely appreciate all participants who agreed to share their data for scientific use and responded to the feedback questionnaire, as well as the healthcare professionals at Checkpoint Zurich and Test-In Zurich, Switzerland, for organising and conducting the visits and contributing to data entry and Raphael Degen (Test-In Zurich) and Lukas Ursprung (Checkpoint Zurich) for the data management at the sites. Further, we would like to thank Hanna Though of the Federal Office of Public Health for her support with the BerDa tool. We gratefully acknowledge the assistance of Céline Capelli in accessing and downloading relevant data from the SwissPrEPared database, facilitating our analysis. We are grateful to the City of Zurich for their continued collaboration and support of the project.

## Author contributions

**Conceptualization:** Leonie Arns-Glaser, Andrea Farnham, Katja Hochstrasser, Jan S. Fehr, Benjamin Hampel.

**Formal analysis:** Leonie Arns-Glaser.

**Supervision:** Benjamin Hampel.

**Visualization:** Leonie Arns-Glaser.

**Writing – original draft:** Leonie Arns-Glaser, Andrea Farnham.

**Writing – review & editing:** Andrea Farnham, Katja Hochstrasser, Jan S. Fehr, Benjamin Hampel.

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
