## [Decision Letter · Decision Letter 0]

21 Oct 2025

Dear Dr. Leonie,

plosone@plos.org. . . . A rebuttal letter that responds to each point raised by the academic editor and reviewer(s). You should upload this letter as a separate file labeled 'Response to Reviewers'.A marked-up copy of your manuscript that highlights changes made to the original version. You should upload this as a separate file labeled 'Revised Manuscript with Track Changes'.An unmarked version of your revised paper without tracked changes. You should upload this as a separate file labeled 'Manuscript'.

We look forward to receiving your revised manuscript.

Kind regards,

Ali Ahmed, PhD

Academic Editor

PLOS ONE

Journal Requirements:

https://journals.plos.org/plosone/s/file?id=ba62/PLOSOne_formatting_sample_title_authors_affiliations.pdf....

“This work was supported by the City of Zurich (https://www.stadt-zuerich.ch/de.html), City Council Resolution GR Nr. 2021/432. Financial support from the Swiss Federal Office of Public Health (FOPH) (https://www.bag.admin.ch/bag/de/home.html) was received by the SwissPrEPared program and study. The funding organisations had neither any role in the design and conduction of the study; nor in the collection, management, analysis and interpretation of the data; nor in the preparation of the manuscript.”

“B.H. reports honoraria for advisory boards, lectures and travel grants paid to himself from the companies Gilead, MSD and ViiV, which are unrelated to the submitted work. J.S.F. reports grants for the institution from Gilead, MSD, and ViiV Healthcare, which are unrelated to the submitted work. The other authors declare no competing interests.”

We note that one or more of the authors are employed by a commercial company: Gilead, MSD, and ViiV Healthcare

“This work was supported by the City of Zurich, City Council Resolution GR Nr. 2021/432. Financial support from the Swiss Federal Office of Public Health (FOPH) was received by the SwissPrEPared program and study. The funding organisations had neither any role in the design and conduction of the study; nor in the collection, management, analysis and interpretation of the data; nor in the preparation of the manuscript.”

“This work was supported by the City of Zurich (https://www.stadt-zuerich.ch/de.html), City Council Resolution GR Nr. 2021/432. Financial support from the Swiss Federal Office of Public Health (FOPH) (https://www.bag.admin.ch/bag/de/home.html) was received by the SwissPrEPared program and study. The funding organisations had neither any role in the design and conduction of the study; nor in the collection, management, analysis and interpretation of the data; nor in the preparation of the manuscript.”

Reviewers' comments:

Reviewer's Responses to Questions

**Comments to the Author**

1. Is the manuscript technically sound, and do the data support the conclusions?

Reviewer #1: Yes

Reviewer #2: Yes

2. Has the statistical analysis been performed appropriately and rigorously?

Reviewer #1: Yes

Reviewer #2: Yes

3. Have the authors made all data underlying the findings in their manuscript fully available?

Reviewer #1: No

Reviewer #2: Yes

4. Is the manuscript presented in an intelligible fashion and written in standard English?

Reviewer #1: Yes

Reviewer #2: Yes

Reviewer #1: This study describes the implementation of free STI counselling and testing for persons < 26 years and people with low income in Zurich, Switzerland. Routinely collected health data and a client feedback questionnaire were used to assess factors associated with first HIV/STI testing in the first 12 months of the program and examined potential barriers. The collected data set appears to be a representative sample of persons using the testing centers. The study is important as the result of this project may give important insights towards implementing free STI counselling and testing in other high income settings where barriers towards testing are different than in low/middle-income settings. The used statistical methods seem appropriate for the study.

My main concern is that research question 3 (impact on workload/capacitiy of the testing centers) cannot be adequately answered with the collected data as this is only evaluated with one question from the voluntary feedback questionnaire on wait times and the mention of the overall performed number of visits. However, there is no information on what the "usual" number of visits is for the testing centers, therefore as a reader no conclusion can be drawn whether this is an increase/additional burden or not. I would suggest to either adapt/remove research question 3 and/or either elaborate more on the limitation of the available data to answer this question or if this data is available include it in the manuscript.

Overall, I would recommend to accept the manuscript with the aboved mentioned major revision.

Additional specific comments:

Background:

1. please include 1-2 sentences on the background of the HIV burden of key populations in Switzerland, so readers know on which populations need to be focused on/reach by testing strategies

Methods:

1. please specify in the methods section that the FBQ questionnaire was not sent to people who were tested during a PrEP visit

Results:

1. Line 159-160: please delete: (Error! Reference source not found.)

2. Table 1: The table in the format I received is hard to read with all the numbers and line changes (often mid-word). Potentially a different format (horizontal), different font size and differing background colors could help to make it more readable and thus understandable. Consider doing two separate tables for characteristics/reasons for accessing VCT and the results

3. Line 167 (+ table): Stage III/laten, please change to Stage III/latent (and a superscripted "e" instead of a "3" as all the other footnotes are letters and not numbers)

4. Is there any information of the educational background in the general population taking up the free testing (not only from the ones completing the voluntary FBQ survey)? This might be interesting to see if in general persons with a higher education are more likely to take up free STI testing or if the bias mainly exists because they are more likely to fill out the survey.

5. Are the people coming for their first STI test the same as coming for their first HIV test or are these different groups? In the persons filling out the FBQ survey the difference between the two groups is only two (with two more first STI testing). I was not able to find the information of the overall study population, in the text only first HIV tests are mentioned. Is there a reason you are not reporting on first STI tests in the overall study population?

Discussion:

- how do you explain the low number of tests during PrEP visits? Different population or is there a different way for subsidized testing

Reviewer #2: 1. In the abstract you have mentioned that "MSM, were significantly less likely to be first-time testers for HIV (OR: 0.28, 95% CI: 0.15–0.48) and STI (OR: 0.74, 95% CI:0.44–1.21) compared to women who have sex with men (WSM)". The confidence interval crosses 1, so the results donot remain significant. Kindly look into it and address this in the abstracts as well as in your manuscript.

2. The BACKGROUND seems not well structured. The starting line "For Switzerland to reach the next World Health Organization (WHO) goal of 95-95-95 in the HIV-cascade by 2025, the reach of its HIV testing programmes need to be expanded" is not impactful. The whole Background needs to be rewritten to create a compelling narrative to capture the reader's interest.

3. The data set counts clinic visits, not named people, and 39% of them were first-time testers. Repeat attenders can bias estimates. Explicitly mentioning "visits as units of analysis" (not individuals) in methodology can clarify the methodology.

A sensitivity analysis where you can assume some plausible repeat rates (like 10%, 20%) and its effect on the main results can also be a valuable addition to the statistical measures.

.

Reviewer #1: No

Reviewer #2: **Yes:** Muhammad Wasay ShahidMuhammad Wasay ShahidMuhammad Wasay ShahidMuhammad Wasay Shahid

---

## [Author Response · Author response to Decision Letter 1]

2 Dec 2025

Dear Editorial Office,

As requested, we have uploaded:

– Response to Reviewers

– Revised manuscript with track changes

– Clean revised manuscript

- Supplementary Information and figures in recommended format

– Updated Funding Statement, Competing Interests, and Data Availability statements (included in the cover letter).

Please let us know if any additional information is required

---

## [Decision Letter · Decision Letter 1]

25 Jan 2026

Dear Dr. Arns-Glaser,

Thank you for submitting your manuscript to PLOS ONE. After careful consideration, we feel that it has merit but does not fully meet PLOS ONE’s publication criteria as it currently stands. Therefore, we invite you to submit a revised version of the manuscript that addresses the points raised during the review process.

We look forward to receiving your revised manuscript.

Kind regards,

Ismael Maatouk, MD, PhD, MPH

Academic Editor

PLOS One

Journal Requirements:

Additional Editor Comments:

Kindly address a minor comment from reviewer 2. Thank you

Reviewers' comments:

Reviewer's Responses to Questions

**Comments to the Author**

Reviewer #1: All comments have been addressed

Reviewer #2: All comments have been addressed

2. Is the manuscript technically sound, and do the data support the conclusions?

Reviewer #1: Yes

Reviewer #2: Yes

3. Has the statistical analysis been performed appropriately and rigorously?

Reviewer #1: Yes

Reviewer #2: Yes

4. Have the authors made all data underlying the findings in their manuscript fully available?

Reviewer #1: Yes

Reviewer #2: Yes

5. Is the manuscript presented in an intelligible fashion and written in standard English?

Reviewer #1: Yes

Reviewer #2: Yes

Reviewer #1: My comments have been adequatly addressed and the manuscript adapted accordingly. I there recommend to accept thr manuscript.

Reviewer #2: You mentioned in the abstract that "MSM were also less likely to be first-time testers for STIs (OR: 0.74, 95% CI: 0.44–1.21) compared to women who have sex with men (WSM)" is not clarified properly. Since, the CI exceeds value of 1, the relation should be reported a "non-significant", "less likely" is rather misleading. Kindly correct it.

.

Reviewer #1: No

Reviewer #2: **Yes:** Muhammad Wasay ShahidMuhammad Wasay ShahidMuhammad Wasay ShahidMuhammad Wasay Shahid

---

## [Author Response · Author response to Decision Letter 2]

5 Mar 2026

Responses to Editors

We would like to thank the editors and reviewers for their careful reassessment of our manuscript and our responses to their comments. Below follow the point-by-point answers to the remaining comments.

Comment 1

Response to comment 1

We thank the editor for the reminder to check our references. In the reference below, there was a wrong DOI number. This has been rectified in the manuscript.

Mitjà O, Padovese V, Folch C, Rossoni I, Marks M, Arias MAR, et al. Epidemiology and determinants of reemerging bacterial sexually transmitted infections (STIs) and emerging STIs in Europe. Lancet Reg Health Eur. 2023;34: 100778. doi:10.1016/j.lanepe.2023.100742

Moreover, the CDC has moved one of the referenced resources. The link has been updated in the reference list:

Centers for Disease Control and Prevention (CDC). Trends in the prevalence of sexual behaviors and HIV testing: National YRBS 1991–2019 [Internet]. 2021 [cited 2026 February 26]. Available from: https://www.cdc.gov/yrbs/?CDC_AAref_Val=https://www.cdc.gov/healthyyouth/data/yrbs/factsheets/2019_sexual_trend_yrbs.htm

Additional Editor Comments:

Kindly address a minor comment from reviewer 2. Thank you

Response to additional Editor Comments

Thank you very much for reminding us of reviewer 2’s comment. We have implemented the change in the abstract to fit with their comment and replied to it below.

Response to Reviewers

We sincerely thank both reviewers for the time and efforts in reassessing our manuscript.

Comment Reviewer 2

You mentioned in the abstract that "MSM were also less likely to be first-time testers for STIs (OR: 0.74, 95% CI: 0.44–1.21) compared to women who have sex with men (WSM)" is not clarified properly. Since, the CI exceeds value of 1, the relation should be reported a "non-significant", "less likely" is rather misleading. Kindly correct it.

Response to Comment Reviewer 2

Thank you very much for pointing out the phrase that could be misunderstood. We now explicitly point out the non-significance of the second association:

“MSM were significantly less likely to be first-time testers for HIV (OR: 0.28, 95% CI: 0.15–0.48) and also less likely to be first-time testers for STIs (OR: 0.74, 95% CI: 0.44–1.21) compared to women who have sex with men (WSM); however, the latter association was non-significant.”

We thank the editors and reviewers once again for their constructive and thoughtful feedback improving our manuscript. We would be grateful for your consideration of the revised version, and we remain available to provide any further information if needed.

Yours sincerely,

Leonie Arns-Glaser

---

## [Decision Letter · Decision Letter 2]

9 Mar 2026

Lowering the barriers to sexual health services: Impacts of free counselling and testing for sexually transmitted infections in Switzerland – an observational study

PONE-D-25-30038R2

Dear Dr. Arns-Glaser,

We’re pleased to inform you that your manuscript has been judged scientifically suitable for publication and will be formally accepted for publication once it meets all outstanding technical requirements.

Kind regards,

Ismael Maatouk, MD, PhD, MPH

Academic Editor

PLOS One

Additional Editor Comments (optional):

Thank you for addressing the reviewer's comments. Congratulations for the acceptance.

Reviewers' comments:

Reviewer's Responses to Questions

**Comments to the Author**

Reviewer #2: All comments have been addressed

2. Is the manuscript technically sound, and do the data support the conclusions?

Reviewer #2: Yes

3. Has the statistical analysis been performed appropriately and rigorously?

Reviewer #2: Yes

4. Have the authors made all data underlying the findings in their manuscript fully available?

Reviewer #2: Yes

5. Is the manuscript presented in an intelligible fashion and written in standard English?

Reviewer #2: Yes

Reviewer #2: (No Response)

.

Reviewer #2: **Yes:** Muhammad Wasay ShahidMuhammad Wasay ShahidMuhammad Wasay ShahidMuhammad Wasay Shahid

---

## [Editor Report · Acceptance letter]

PONE-D-25-30038R2

PLOS One

Dear Dr. Arns-Glaser,

I'm pleased to inform you that your manuscript has been deemed suitable for publication in PLOS One. Congratulations! Your manuscript is now being handed over to our production team.

Kind regards,

on behalf of

Dr. Ismael Maatouk

Academic Editor

PLOS One